# Pregnancy Associated Plasma Protein-A as a Cardiovascular Risk Marker in Patients with Stable Coronary Heart Disease During 10 Years Follow-Up—A CLARICOR Trial Sub-Study

**DOI:** 10.3390/jcm9010265

**Published:** 2020-01-18

**Authors:** Erik Nilsson, Jens Kastrup, Ahmad Sajadieh, Gorm Boje Jensen, Erik Kjøller, Hans Jørn Kolmos, Jonas Wuopio, Christoph Nowak, Anders Larsson, Janus Christian Jakobsen, Per Winkel, Christian Gluud, Kasper K Iversen, Johan Ärnlöv, Axel C. Carlsson

**Affiliations:** 1Department of Medical Epidemiology and Biostatistics, Karolinska Institutet, 17177 Stockholm, Sweden; 2School of Medical Sciences, Örebro University, 70182 Örebro, Sweden; 3Department of Cardiology, Rigshospitalet University of Copenhagen, 2100 Copenhagen, Denmark; jens.kastrup@regionh.dk; 4Department of Cardiology, Copenhagen University Hospital of Bispebjerg and Frederiksberg, 2000 Frederiksberg, Denmark; ahmad.sajadieh@regionh.dk; 5Department of Cardiology, Hvidovre Hospital University of Copenhagen, 2650 Hvidovre, Denmark; gorm.boje.jensen.01@regionh.dk; 6Department of Cardiology S, Herlev Hospital University of Copenhagen, 2730 Herlev, Denmark; kjoller@dadlnet.dk (E.K.); kasper.karmark.iversen@regionh.dk (K.K.I.); 7Department of Clinical Microbiology, Odense University Hospital, 5000 Odense, Denmark; h.j.kolmos@dadlnet.dk; 8Department of Medicine, Mora County Hospital, 79251 Mora, Sweden; jonas.wuopio@ltdalarna.se; 9Division for Family Medicine and Primary Care, Department of Neurobiology, Care Sciences and Society, Karolinska Institutet, 14183 Huddinge, Sweden; christoph.nowak@ki.se (C.N.); axelcefam@hotmail.com (A.C.C.); 10Department of Medical Sciences, Uppsala University, 75185 Uppsala, Sweden; anders.larsson@akademiska.se; 11Copenhagen Trial Unit, Centre for Clinical Intervention Research, Rigshospitalet, Copenhagen University Hospital, 2100 Copenhagen, Denmark; janus.jakobsen@ctu.dk (J.C.J.); per.winkel@ctu.dk (P.W.); christian.gluud@ctu.dk (C.G.); 12Department of Cardiology, Holbæk Hospital, 4300 Holbæk, Denmark; 13School of Health and Social Studies, Dalarna University, 79131 Falun, Sweden; johan.arnlov@ki.se

**Keywords:** pregnancy-associated plasma protein-A, coronary artery disease, cohort studies, biomarkers

## Abstract

Elevated pregnancy-associated plasma protein A (PAPP-A) is associated with mortality in acute coronary syndromes. Few studies have assessed PAPP-A in stable coronary artery disease (CAD) and results are conflicting. We assessed the 10-year prognostic relevance of PAPP-A levels in stable CAD. The CLARICOR trial was a randomized controlled clinical trial including outpatients with stable CAD, randomized to clarithromycin versus placebo. The placebo group constituted our discovery cohort (*n* = 1.996) and the clarithromycin group the replication cohort (*n* = 1.975). The composite primary outcome was first occurrence of cardiovascular event or death. In the discovery cohort, incidence rates (IR) for the composite outcome were higher in those with elevated PAPP-A (IR 12.72, 95% Confidence Interval (CI) 11.0–14.7 events/100 years) compared to lower PAPP-A (IR 8.78, 8.25–9.34), with comparable results in the replication cohort. Elevated PAPP-A was associated with increased risk of the composite outcome in both cohorts (discovery Hazard Ratio (HR) 1.45, 95% CI 1.24–1.70; replication HR 1.29, 95% CI 1.10–1.52). In models adjusted for established risk factors, these trends were attenuated. Elevated PAPP-A was associated with higher all-cause mortality in both cohorts. We conclude that elevated PAPP-A levels are associated with increased long-term mortality in stable CAD, but do not improve long-term prediction of death or cardiovascular events when added to established predictors.

## 1. Introduction

Pregnancy-associated plasma protein-A (PAPP-A) is a cell membrane-bound metalloproteinase which regulates local availability of insulin-like growth factor 1 (IGF-1) [1]. It has been evaluated as a prognostic biomarker in acute coronary syndromes [2,3,4,5,6,7,8,9,10,11], where elevated levels are associated with increased risk of death. However, this association may be confounded by heparin treatment causing elevated PAPP-A levels in vivo through release of PAPP-A attached to cell membranes [12]. Further, PAPP-A levels predict cardiovascular events in troponin-negative patients with suspected acute coronary syndrome [3] and higher levels of circulating PAPP-A are associated with more extensive coronary artery disease [13] as well as with plaque inflammation and echogenicity [14]. In chronic stable angina pectoris PAPP-A has been less extensively studied and is associated with outcomes in some studies [15,16,17,18] but not all [19]. Studies with long-term follow-up are scarce.

PAPP-A regulates downstream growth hormone effects in the paracellular environment by cleaving insulin-like growth factor binding protein 4 (IGFBP4) bound to insulin-like growth factor 1 (IGF-1), thereby making IGF-1 available to its receptor. Since PAPP-A is normally bound to the cell membrane and not abundantly expressed, increased plasma levels in the absence of heparin treatment may represent up-regulation due to inflammation or tissue damage, in combination with escape into the circulation [20,21,22].

The present study is part of the larger PREdictors for MAjor Cardiovascular outcomes in stable ischemic heart disease (PREMAC) study, which aimed to identify biochemical predictors of cardiovascular events and all-cause mortality in persons with stable coronary artery disease (CAD) utilizing data originating from the CLARICOR (clarithromycin for patients with stable coronary heart disease) trial [23]. In the CLARICOR Trial, patients were randomized to clarithromycin or placebo and the main outcomes consisted of myocardial infarction (AMI), unstable angina pectoris (UAP), cardiovascular mortality, and all-cause mortality. In a previous report on the same cohort, elevated PAPP-A, defined as values ≥ 4 µ/mL, was found to predict risk of death and myocardial infarction during medium term (median 2.8 years) follow up [17]. The aim of the present study was to assess the predictive power of elevated PAPP-A levels for the 10-year outcomes in stable CAD.

## 2. Experimental Section

The PREMAC study focused on the presence of predictors of cardiovascular events and all-cause mortality in persons with stable CAD and included a detailed statistical analysis plan [23]. Biomarker assessment was performed using stored biobank samples from the CLARICOR trial [24] and outcome data was retrieved from public registers. The CLARICOR trial was approved by local ethics committees and regulatory authorities (Regional Ethics Committee HB 2009/015 and KF 01-076/99; the Danish Data Protection Agency 1999–1200–174 and 2012–41–0757; and the Danish Medicines Agency 2612–975).

### 2.1. Patient Selection

The CLARICOR trial was a randomized, placebo-controlled trial with blinded outcome assessment including outpatients with stable CAD. All patients discharged from wards or outpatient clinics in the Copenhagen area in Denmark with a diagnosis of acute myocardial infarction or unstable angina pectoris during the years 1993–1999 who were alive and aged 18–85 years old in 1999 (*n* = 13.702) were invited to a screening interview at one of five cardiology centers. Of the 6116 (44.6%) patients accepting the invitation, 1567 (25.6%) were excluded, 177 (2.9%) chose not to participate, and the remaining 4372 (71.5%) were randomized to oral clarithromycin 500 mg once daily for 2 weeks (*n* = 2.172) vs. placebo (*n* = 2.200) during the winter 1999–2000. Exclusion criteria of the CLARICOR trial were: AMI or UAP within the previous 3 months, percutaneous transluminal coronary angioplasty and coronary bypass surgery within the previous 6 months, impaired renal or hepatic function, congestive heart failure (New York Heart Association (NYHA) IV classification of heart failure), active malignancy, incapacity to manage own affairs, breast feeding, and possible pregnancy. In the CLARICOR trial, clarithromycin was found to increase both the risk of cardiovascular and all-cause mortality [24,25,26,27].

The patients randomized to placebo in the CLARICOR study were included as the discovery cohort in the present study, while those randomized to clarithromycin formed the replication cohort. We excluded participants with missing data in any of the variables, leaving *n* = 1.996 (92%) in the discovery cohort, and *n* = 1.975 (90%) in the replication cohort.

### 2.2. Baseline Data

During enrollment interviews, smoking status, current medication, and known hypertension or diabetes were noted. Information concerning sex, age, and history of myocardial infarction or unstable angina pectoris were extracted from local hospital files. Blood samples were collected at each of the study sites immediately before randomization, using blood collection tubes without additives. Serum was prepared according to normal hospital routine with approximately coagulation for 30 min and centrifugation at 1500 *g* for 10 min. Serum was frozen on the day of collection at −20 °C and at −80 °C after transportation to the central laboratory facility. Storage problems were the only noteworthy cause of missing data. Estimated glomerular filtration rate (eGFR) was calculated using the creatinine-based Chronic Kidney Disease Epidemiology Collaboration (CKD-EPI) formula [28]. Smoking status was categorized as never, former, or current smoker. No physical investigations were made at randomization interview; nor were any longitudinal predictor information collected during follow-up.

### 2.3. Pregnancy-Associated Plasma Protein A Levels

The PAPP-A levels measured in a previous study were used in the present study [17]. The enzyme-linked immunosorbent assay used for quantification of PAPP-A has been described in detail previously [17,29]. The detection limit was 4 mIU/L. The intra-assay coefficient of variation was 2.0% at 71.7 mIU/L and 5.7% at 10.4 mIU/L, with corresponding inter-assay coefficients of variation of 6.4% and 8.7%, respectively. Elevated serum PAPP-A was defined as values at or above 4 mIU/L, based on levels in healthy blood donors [29]. Note that although the CLARICOR trial data did not include information on heparin use, study participants were outpatients with stable CAD and heparin is not used in this setting.

### 2.4. Outcomes

Follow-up was until 31 December 2009 where the official permissions expired. Outcome data was procured from national patient registries. These are mandatory for inpatient care and all events diagnosed and coded during hospital admission are therefore detected, resulting in virtually no loss to follow-up. Vital status was retrieved from the Danish Central Civil Register, cause of death from the National Register of Causes of Death, and hospital admissions from the Danish National Patient Register (NPR), which covers all hospital admissions. These registries have almost complete coverage [30]. By trial protocol, events during the first 2.6 years of follow-up were adjudicated by a blinded committee, previously described in detail [23,24]. For the 10-year studies, registry outcomes were used after verifying that the results were consistent with those based on adjudication data [30,31].

The Danish 10-digit central person registration (CPR) number is used at all contacts with the health care system. At discharge from hospital, at least one action diagnosis (A diagnosis) specifying the main reason for the admission is noted in the NPR. These A diagnoses, and in case of death the ‘underlying cause of death’ code (in the official terminology of the National Register of Causes of Death), was used for classifying outcomes according to the 10th revision of the International Statistical Classification of Diseases and Related Health Problems (ICD-10) coding system as follows: AMI (I21.0–23.9), UAP (I20.0 and I24.8–24.9), cerebrovascular disease (CeVD) (I60.0–64.9 and G45.0–46.8), cardiovascular death (I00.0–99.9 unless already covered), and death due to non-cardiovascular disease (A00.0–T98.3 unless already covered). A composite outcome was defined as AMI, UAP, CeVD, or death due to any cause. Follow-up time was censored at occurrence of an outcome, death, or end of follow-up (31st December 2009 giving a median possible survival time of 10 years ± 3 months after randomization). 

### 2.5. Statistical Analysis

Incidence rates (IR) were calculated using only the first occurrence of an event and the time to event or censoring at end of study was used in the denominator. We used Cox proportional hazards model for the statistical analysis. Multivariable models were adjusted according to the pre-specified analysis plan, for clinical predictors (sex, age at randomization, smoking history, history of myocardial infarction, hypertension, and diabetes), medical treatment (acetylsalicylic acid, beta-blocker, calcium-antagonist, angiotensin-converting enzyme (ACE)-inhibitor, long lasting nitrate, diuretic, digoxin, statin, and anti-arrhythmic drugs), and standard biochemical predictors (log-transformed high-sensitivity-reactive protein (CRP), glomerular filtration rate (GFR) estimated by creatinine, triglycerides, total cholesterol, high-density lipoprotein (HDL) cholesterol, low-density lipoprotein (LDL) cholesterol, apolipoprotein A1, and apolipoprotein B). Standard predictors adjusted for in multivariable models are listed in Appendix B. Triglycerides and total cholesterol were log transformed.

As the proportional hazard’s assumption was violated for age at entry for all-cause death and the composite outcome (Bonferroni adjusted *p* < 0.0044 for all-cause mortality; and *p* < 0.00056 for the composite outcome), we excluded age from all models for these two outcomes. In order to provide additional insights into the potential influence of age on these associations, we conducted multivariable logistic regression models (including age as a co-variate since the proportional hazard assumption is not a requisite for these analyses).

## 3. Results

Baseline characteristics of the discovery and replication cohorts are presented in Table 1 They showed no major differences between the cohorts. The proportion of participants with elevated PAPP-A levels was 13% (*n* = 263) in the discovery cohort and 12% (*n* = 244) in the replication cohort.

Table 2 displays outcomes by PAPP-A level. In the discovery cohort, the composite outcome was more common among those who elevated PAPP-A, compared to those with low PAPP-A levels (72% compared to 59%), with a corresponding difference in incidence rates (*p* < 0.0001). The same pattern was seen in the replication cohort.

In the survival analysis (Table 3), Cox proportional hazards models adjusted for sex (model A) showed that PAPP-A ≥ 4 µ/mL was associated with an increased risk of the composite outcome in the discovery cohort and, less markedly, in the replication cohort; with adjustment for a large number of other risk factors (model B, see Table 3), comprising comorbidities and biochemical markers, these risk trends were attenuated in the discovery cohort and disappeared in the replication cohort.

Comparable results, with elevated PAPP-A being associated with increased risk of the composite outcome in the discovery cohort, but not in the replication cohort, were found in adjusted logistic regression models (Appendix A) where age was also included in the multivariable model. We found no interaction between sex and PAPP-A on mortality (*p* = 0.22) or on the composite outcome (*p* = 0.17).

The association between PAPP-A ≥ 4 mIU/L and other outcomes is shown in Appendix A. Elevated PAPP-A was associated with higher all-cause mortality in the discovery cohort, an association that remained in the fully adjusted analysis as well as in a logistic regression that included age as a predictor variable (Appendix A). These findings were reproduced in the replication cohort.

Interestingly, these secondary analyses suggest that PAPP-A elevation is at least as strong a predictor for non-cardiovascular as for cardiovascular death (Appendix A). 

We also evaluated the predictive utility of PAPP-A in the group of placebo-treated patients when added to a large number of standard predictors (Appendix A). Adding elevated PAPP-A as a predictor produced no visible improvements (apart from a slight numerical instability).

## 4. Discussion

Our main finding is that PAPP-A levels ≥ 4 mIU/L are associated with increased long-term risk of composite adverse outcome as well as all-cause mortality in patients with stable CAD. Although the association to all-cause mortality remained after extensive multivariable adjustment, the association to the composite outcome was not reproduced in our replication cohort when adjusted for many other risk factors. This may indicate that the placebo group and the clarithromycin treated group (the replication group) are not completely compatible in that clarithromycin was found to increase mortality [24,25,26,27]. It may also indicate that the association between PAPP-A and outcomes is confounded by some of these other risk factors, for example diabetes [20,32]. Over-adjustment may also be a problem in this context [33]. Our choice of covariates in the various multivariate analyses was mandated by the choice made for the Cox analyses of the placebo-treated patients described in the pre-specified analysis plan. 

The growth hormone (GH) axis and PAPP-A has been implicated in the progression of atherosclerosis. Locally, insulin-like growth factor 1 (IGF-1) promotes multiple mechanisms involved in plaque formation and an association between PAPP-A and atherosclerosis has been demonstrated [2,34]. PAPP-A is found in atherosclerotic plaques on cell-types involved in the atherosclerotic process, including vascular smooth muscle cells, endothelial cells and macrophages [1], its expression is elevated in vulnerable plaques [2]. PAPP-A activity is related to atherosclerotic lesion size in rodents [35,36] and higher levels of circulating PAPP-A are associated with more extensive coronary artery disease in humans [13].

Reduced PAPP-A activity is associated with diminished vascular cell proliferation in response to injury, reduced plaque area and less luminal occlusion in atherosclerosis [37]. Conversely, increased PAPP-A activity is associated with proliferation of vascular smooth muscle cells [34]. Regulation of vascular smooth muscle cell proliferation could therefore be a mechanism by which PAPP-A influences the atherosclerotic process. PAPP-A could also be linked to atherosclerosis through modulating the effects of IGF-1 on lipid-, glucose-, and protein metabolism [38,39].

However, the exact mechanism by which PAPP-A is involved (causally or as a by-product) in the promotion of atherosclerosis remains elusive, in part as a result of conflicting findings on the effects of IGF-1 action on the vasculature [40]. Notably, the relationship between serum levels of IGF-1 and PAPP-A and local IGF-1 activity is unclear and may for example be dependent on body composition, inflammation, or conditions such as diabetes mellitus or obesity [40]. Circulating levels are therefore not necessarily directly related to the hypothesized pathophysiological mechanisms by which IGF-1 and PAPP-A are implicated in development of coronary artery disease. Consequently, the association between PAPP-A and mortality described in the present study may be related to other factors than cardiovascular disease progression. Indeed, we did not find any clear association with cardiovascular outcomes and it should be noted that the IGF-1 system including PAPP-A may for example be related to development of cancer [41]. There was in our results consistently no association between elevated PAPP-A and myocardial infarction, UAP, or stroke, although there was an association to cardiovascular mortality in the minimally adjusted analysis in the replication cohort.

The association between PAPP-A levels and outcomes in stable CAD has been studied previously. In 103 stable CAD patients, with a median follow-up of 4.9 years, higher PAPP-A was associated with increased mortality as well as the composite outcome of death and acute coronary syndrome [15]. Interestingly, but potentially problematic [33], those authors adjusted their estimates for the extent of coronary atherosclerosis, which could be considered an intermediate in the hypothesized causal pathway between PAPP-A and cardiovascular events. In another cohort study, including 534 patients with stable CAD and 393 patients with acute coronary syndrome, with a median follow-up time of 5.0 years, the authors found no association to cardiovascular mortality in the subgroup stable CAD, but higher PAPP-A was associated with increased cardiovascular mortality in the overall cohort as well as in the acute coronary syndrome (ACS) subgroup [19]. Although these results were adjusted for several conventional predictors, there was no adjustment for age. In a previous study on the CLARICOR cohort participants, PAPP-A levels were studied in relation to medium-term outcomes [17]. Important differences in that study from the present study include a shorter follow-up (median 2.8 years), joining of the placebo and the treatment group in a single cohort, and differing definitions of the composite outcome. In line with our present results, the previous study found that elevated PAPP-A was associated with the composite outcome of myocardial infarction and death as well as all-cause mortality in adjusted analyses [17].

PAPP-A has also been studied in ACS, but there is limited generalizability from these studies to the context of stable CAD. In troponin-negative patients with suspected ACS, PAPP-A predicted future cardiovascular events [3], although we are uncertain if PAPP-A was sampled before administration of heparin. Others found that PAPP-A predicted cardiovascular events in ACS and it seems that PAPP-A was sampled after heparin infusion, indicating that the PAPP-A levels included PAPP-A released from the cell membrane during heparin treatment [6]. In that study, PAPP-A predicted outcomes in TnT-negative patients. Furthermore, differences in stable versus acute CAD may have support in PAPP-A physiology since chronic and transient PAPP-A expression may have differing effects on neointimal formation following vascular injury [42].

Our present study has several strengths: a large study sample, detailed characterization of the participants, longitudinal study design, 10 years follow-up, and a replication of all analyses in the clarithromycin group of the trial. As far as we know, there are no other large cohort studies on associations between PAPP-A levels in patients with stable CAD. National Danish registers are known to be of high completeness and accuracy [30], but a small number of non-fatal events can be missed when participants are hospitalized abroad. Results in our study are likely valid for patients with stable CAD as ascertained at the baseline interview and it remains to be shown if similar risks are seen for other relevant patient groups, such as patients with acute symptoms or patients during recovery from a major event.

Limitations are the unknown generalizability to other ethnic groups and to those unlikely to volunteer to participate in studies. Distortion by the active intervention with clarithromycin cannot be excluded, although we saw similar associations as to those in the placebo cohort. As regards the replication cohort, with its previously described surplus of unfavorable cardiovascular outcomes [26,27,43], we noted that elevated PAPP-A here tended to lose its unfavorable implications. Such interaction, if present, would imply that the harmful effect of clarithromycin was more marked in those with low PAPP-A levels. However, the trend nowhere came close to statistical significance. Nor do we have any theoretical arguments in favor thereof. Another limitation is that there was no data on heparin treatment at baseline. However, as the participants in our study had no indication for heparin, it is unlikely that this lack of data would have any substantial influence on our results and conclusions. In addition, there was no data on left ventricular ejection fraction, although this may be partially or completely compensated by other covariables included in the analyses, as age, sex, hypertension, prior acute myocardial infarction, creatinine, diuretics, and digoxin are related to left ventricular ejection fraction [44].

## 5. Conclusions

Elevated PAPP-A levels are associated with increased long-term mortality in stable CAD, but they do not improve long-term prediction of composite outcome of death or cardiovascular events when added to established predictors.

## Figures and Tables

**Table 1 jcm-09-00265-t001:** Baseline characteristics of the two study cohorts.

Variable	Discovery Cohort	Replication Cohort
Number of participants	1996	1975
PAPP-A ≥ 4 mIU/L	263 (13)	244 (12)
Female	623 (31)	602 (30)
Age at entry, years	65 ± 10	65 ± 10
CRP, mg/L	5.25 ± 7.7	5.76 ± 9.3
Apolipoprotein A1, mg/dL	1.70 ± 0.34	1.70 ± 0.36
Apolipoprotein, mg/dL	1.21 ± 0.32	1.21 ± 0.33
eGFR, mL/min	76.3 ± 20	76.5 ± 19
Diabetes mellitus	299 (15)	301 (15)
Hypertension	805 (40)	790 (40)
Never smoked	394 (20)	339 (17)
Former smoker	925 (46)	903 (46)
Current smoker	677 (34)	735 (37)
History of myocardial infarction	635 (32)	640 (32)
Statin treatment	822 (41)	812 (41)
Aspirin treatment	1763 (88)	1733 (88)
Beta blocker treatment	619 (31)	589 (30)
Calcium antagonist treatment	702 (35)	680 (34)
ACE inhibitor treatment	522 (26)	552 (28)
Long-acting nitrate treatment	412 (21)	411 (21)
Diuretics treatment	690 (35)	698 (35)
Digoxin treatment	115 (6)	138 (7)
Antiarrhythmic treatment	42 (2)	46 (2)

Baseline characteristics in the discovery (placebo) and replication (clarithromycin) cohorts, presented as mean ± standard deviation for continuous variables and *n* (%) for categorical variables. Abbreviations: PAPP-A: pregnancy-associated plasma protein A; CRP: high sensitivity C-reactive protein; eGFR: estimated glomerular filtration rate; ACE: angiotensin converting enzyme.

**Table 2 jcm-09-00265-t002:** Incidence rates of the composite outcome by PAPP-A level.

PAPP-A Category	Variable	Discovery Cohort	Replication Cohort
PAPP-A ≥ 4 mIU/L	*N*	263	244
	Outcomes, *N* (%)	189 (72)	168 (69)
	IR, per 100 years	12.72	12.04
	95% CI	11.0–14.7	10.35–14.01
PAPP-A < 4 mIU/L	*N*	1733	1731
	Outcomes, *N* (%)	1015 (59)	1052 (61)
	IR per 100 years	8.78	9.38
	95% CI	8.25–9.34	8.83–9.96

The composite outcome was defined as acute myocardial infarction, unstable angina pectoris, cerebrovascular disease, or death due to any cause. Outcomes (with % of participants at risk) is the number of persons experiencing the composite outcome during 10-years follow-up. Incidence rates (IR) were calculated using only the first occurrence of the outcome during follow-up. Abbreviations: PAPP-A: pregnancy-associated plasma protein A; CI, Confidence Interval.

**Table 3 jcm-09-00265-t003:** Risk of composite outcome associated with the binary covariate elevated PAPP-A.

Variable	Discovery Cohort		Replication Cohort	
	Model A	Model B	Model A	Model B
Hazard ratio	1.45	1.29	1.28	1.06
95% CI	1.24–1.70	1.10–1.52	1.08–1.50	0.89–1.25
*p*-value	< 0.001	< 0.001	0.003	0.51

Cox proportional hazards models are applied to the composite outcome defined as acute myocardial infarction, unstable angina pectoris, cerebrovascular disease, or death due to any cause. Model A was adjusted for sex. Model B was adjusted for established risk factors and co-morbidities, standard biochemical predictors, and treatments as listed in Appendix B. All models in this table are shown without adjustments for age at entry. Abbreviations: PAPP-A, pregnancy-associated plasma protein A.

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
