# Peer review of "Pregnancy Associated Plasma Protein-A as a Cardiovascular Risk Marker in Patients with Stable Coronary Heart Disease During 10 Years Follow-Up—A CLARICOR Trial Sub-Study"

_jcm, 2020, doi:10.3390/jcm9010265_

Round 1
Reviewer 1 Report
Summary: Elevated levels of PAPP-A are associated with increased risk of death in acute coronary syndromes PAPP-A levels predict cardiovascular events in troponin-negative patiens with suspected acute coronary syndrome The is part of PREMAC study wich aimed to identify biochemical predictors of cardiovascular events and all-cause mortality in persons with stable coronary artry disease utilizing data originating from the CLARICON study In a previous report on the same cohort, elevated PAPP-A was found to predict risk of death and myocardial infarction during medim term follow-up Aim of the study is to assess the predictive power of elevated PAPP-A levels for the 10 years outcomes in stable CAD The patiens randomized to placebo in the CLARICOR study were included as the discovery chort in the present study Incidence rates were calculated using only the first occorrence of an event The main finding is that PAPP-A levels ≥ 4 mIU/L are associated with increased long-term risk of composite adverse outcomes as well as all-cause mortality in patiens with stabe CAD. These results were not found in the clarithromycin group The authors conclude that the association between PAPP-A and outcomes in confounded by some other risk factors The discussion analyzes the possible role of PAPP-A and GH in the progression of atherosclerosis and in my opinion this role could be better defined although it is correctly pointed out that some pathophysiological mechanisms are still undefined or conflicting in the conclusions Consideration PAPP-A was detected 40 years ago in the plasma of pregnant women and was immediately suspected to have several biological roles due to similarities to other carrier and modulator proteins. PAPP-A is a high-molecular-mass glycoprotein, which is mainly synthesized by the syncytiotrophoblast and has originally been proposed for the screening of Down syndrome PAPP-A is one of the two pappalysins, a protease that belongs to the superfamily of metzincin (such as astacins, adamalysins, serralysins, the matrix metalloproteinases, and the pappalysins) of metalloproteinases and participates in the local release of insulin-like growth factors. PAPP-A consists of a heterotetrameric disulfide bound 2:2 complex composed of two 200 kDa PAPP-A subunits the 206-residue proform of eosinophil major basic protein (proMBP) subunits and has been assigned to human chromosome 9q 33.1. ProMBP functions as a proteinase inhibitor of PAPP-A. During pregnancy, these two glycosylated subunits are produced separately and then united secondarily. Although the PAPP-A mRNA can be found in many tissues, the placental production exceeds any other. The syncytiotrophoblast composes the PAPP-A subunit, while proMBP is synthesized in extravillous trophoblasts, and the two subunits are combined to create the final complex at the extracellular environment The role of insulin-like growth factor-I (IGF-I) and IGF-II is regulated by binding to one of six IGF binding proteins (IGFBP). PAPP-A is a highly selective enzyme and the responsible proteinase for cleavage of IGF binding proteins (specifically, IGFBP-4, IGFBP-5, and to some extent, IGFBP-2), acting as a regulator of IGF’s bioavailability. PAPP-A2, the second member of the pappalysin family, cleaves only IGFBPs 3 and 5. IGFBP is proteolyzed by PAPP-A, the bioavailability of IGF is enhanced, trophoblast invasion is mediated, and glucose and amino acids transport in the placenta is determined. PAPP-A is proven to be highly increased in blood serum of up to 100 and 10,000 times in the first and third trimesters of pregnancy, respectively, compared to the samples of nonpregnant women To date, there is no evidence of substrates, other than IGFBPs, that PAPP-A reacts with. However, except cleavaging insulin-like growth factors, several other studies about the biological role of PAPP-A have demonstrated a possible complementary role. Atherosclerosis was first to be correlated to increased PAPP-A in the 2001 study by Bayes-Genis et al. and since then, several studies aimed at clarifying the connection between coronary syndrome and other atherosclerotic diseases to this metalloproteinase. Furthermore, increased levels of glomerular PAPP-A revealed an association with diabetic neuropathy, speculating a potential therapeutic role on inhibiting PAPP-A. Moreover, the involvement of PAPP-A in tumor development has been increasingly supported with an expression of PAPP-A in ovarian, breast, lung, and other cancer tissues constantly being reported In 2001, Bayes-Genis et al first suggested that PAPP-A may be a promising candidate biomarker of ischemic heart disease, reporting an AUC of 0.94 for discriminating patients with acute myocardial infarction (AMI) from those without ischemic heart injury Since major controversy still exists on the clinical usefulness of PAPP-A for the early diagnosis of AMI and the results of a meta-analysis including 2050 patients with ischemic heart disease suggest that PAPP-A may not be useful in the early diagnosis of AMI PAPP-A levels are significantly elevated in CKD patients especially in those with worse cardiovascular outcome Conclusion The manuscript is well written and well organised. It describe a possible role of biomarkers PPA_A in the stratification of cardiovascular risk. It is interesting to read and the the conclusions are supported by the analysis of the data presented My suggestions are: expand and try to hypothesize the possible role of PAPP-A in the progression of atherosclerosis in order to justify the finding of the increase in mortality found in the group studied or, alternatively, consider the increase in PAPP-A only an occasional and confounding finding related to association with other risk factors such as diabetes or CKD
Author Response
We thank the reviewer for the kind remarks, the excellent summary of our findings and for expanding on heterotetrameric PAPP-A and outlining the role of cell membrane bound PAPP-A in the GH/IGF-1 system. We do agree that it would be very interesting to achieve better insights into why the circulating levels of PAPP-A increase the risk of mortality in our study sample. Unfortunately, our observational study design precludes any firm conclusions regarding the underlying mechanisms of our findings and any mechanistic hypothesis based on our data should be considered speculative. Additional experimental studies are needed to address the issue of causal mechanisms. Based on this inherent shortcoming of all observational studies, we are reluctant to add too much speculation on underlying mechanisms. Still, in accordance with the reviewers suggestion, we have added some new content to the discussion on the role of PAPP-A in progression of atherosclerosis and included an additional reference (Steffensen 2019) on this in the introduction section (Discussion, page 6):
“The growth hormone (GH) axis and PAPP-A has been implicated in the progression of atherosclerosis. Locally, insulin-like growth factor 1 (IGF-1) promotes multiple mechanisms involved in plaque formation and an association between PAPP-A and atherosclerosis has been demonstrated [2,34]. PAPP-A is found in atherosclerotic plaques on cell-types involved in the atherosclerotic process, including VSMCs, endothelial cells and macrophages [1], its expression is elevated in vulnerable plaques [2]. PAPP-A activity is related to atherosclerotic lesion size in rodents [35,36] and higher levels of circulating PAPP-A are associated with more extensive coronary artery disease in humans [13].
Reduced PAPP-A activity is associated with diminished vascular cell proliferation in response to injury, reduced plaque area and less luminal occlusion in atherosclerosis [37]. Conversely, increased PAPP-A activity is associated with proliferation of vascular smooth muscle cells [34]. Regulation of vascular smooth muscle cell proliferation could therefore be a mechanism by which PAPP-A influences the atherosclerotic process. PAPP-A could also be linked to aterosclerosis through modulating the effects of IGF-1 on lipid-, glucose- and protein metabolism [38,39].”
Reviewer 2 Report
This manuscript extends follow-up time to 10 years from an average of 2.8 years in a previously published study by this group to assess circulating PAPP-A levels as a predictive biomarker for cardiovascular risk in patients with stable coronary artery disease (CAD). Strenghts were the large sample size, detailed characterization of patients with stable CAD, and experience with PAPP-A assays.
Specific Comments:
The authors point out the confounding effects of heparin treatment, so it should be clearly stated that heparin was not used in any of these patients. Blood samples were drawn and stored at -80C. How were the blood samples collected and processed? Serum, plasma? This information was not in their 2011 paper either and should be included. It is important that there is consistency. Minor but potentially misleading statement at line 59 - "PAPP-A regulates growth hormone action in the paracellular environment...". growth hormone should be replaced by IGF.
Author Response
Since the reviewer had more than one suggestion, point-by-point answers are provided in the attached file.

Round 2
Reviewer 2 Report
The authors have adequately addressed my concerns